# Procalcitonin for Early Detection of Pharyngocutaneous Fistula after Total Laryngectomy: A Pilot Study

**DOI:** 10.3390/cancers16040768

**Published:** 2024-02-13

**Authors:** Massimo Mesolella, Salvatore Allosso, Gerardo Petruzzi, Antonietta Evangelista, Giovanni Motta, Gaetano Motta

**Affiliations:** 1Unit of Otorhinolaryngology, Department of Neuroscience, Reproductive Sciences and Dentistry, University Federico II of Naples, 80131 Naples, Italy; 2Department of Otolaryngology—Head and Neck Surgery, IRCCS Regina Elena National Cancer Institute, 00144 Rome, Italy; petruzzigerardo@gmail.com; 3Unit of Otorhinolaryngology—Santa Scolastica Hospital, 03043 Cassino, Italy; a.evangelista919@gmail.com; 4Unit of Otorhinolaryngology, Department of Mental and Physical Health and Preventive Medicine, University Luigi Vanvitelli, 80131 Naples, Italy; giovannimotta95@yahoo.com (G.M.); gaetano.motta@unicampania.it (G.M.)

**Keywords:** procalcitonin, pharyngocutaneous fistula, biomarker, surgery complication, head and neck cancer, laryngeal cancer

## Abstract

**Simple Summary:**

The purpose of our study was to evaluate the effectiveness of procalcitonin (PCT) as a positive predictive factor for the early identification of post-surgical wound complications, such as cutaneous pharyngeal fistulas, in patients undergoing total laryngectomy. We took into consideration 36 patients who underwent total laryngectomy, dividing them into two groups: a first group made up of 27 patients who had no post-operative complications; a second group made up of 9 patients who had the onset of pharyngeal-cutaneous fistula as a complication. In both groups we evaluated the procalcitonin values at various times in order to evaluate the values found with the onset of the complication.

**Abstract:**

Objectives. The aim of this prospective study was to investigate the role of procalcitonin as an early diagnostic marker of pharyngocutaneous fistula (PCF) in a cohort of head and neck patients treated with total laryngectomy for squamous cell carcinoma. Methods. This prospective study was conducted on a sample of patients enrolled from January 2019 to March 2022. All patients were subjected to a “protocol” of blood chemistry investigations, scheduled as follows: complete blood count with formula, ESR dosage, CPR, and PCT. PCT was also dosed by salivary sampling and a pharyngo-cutaneous swab in patients who presented with PCF. The dosage scheme was systematically repeated: the day before the intervention (t0); the 5th day postoperative (t1); the 20th day postoperative (t2); and at time X, the day of the eventual appearance of the pharyngocutaneous fistula. Results. A total of 36 patients met the inclusion criteria. The patients enrolled in the study were subsequently divided into two groups: 27 patients underwent total laryngectomy (TL) for laryngeal cancer without postoperative complications, and 9 patients were undergoing TL with postoperative PCF. Using the Cochran’s Q test, statistical significance was found for PCT among T0, T1, Tx, and T2 (*p*-value < 0.001) between the PCF and non-PCF groups. The Z test demonstrated that there is a difference in PCT levels at T1 and T2 and that this difference is statistically significant (*p* < 0.001). Conclusions. PCT could be considered an early marker of complications in open laryngeal surgery. According to our results, it could be useful in the precocious detection of pharyngocutaneous fistulas and in the management of antibiotic therapy.

## 1. Introduction

The increasing prevalence of antibiotic resistance is a large-scale emergency. It is reported that treatment indications, i.e., doses and durations of treatments, are incorrect in up to 30–50% of antibiotic prescriptions [1,2,3,4]. This may be associated with increased costs, adverse events, and the prolonged duration of hospitalization. On the other hand, early prescription may be necessary to avoid severe bacterial illness [5].

In the past 10 years, the infection blood biomarker procalcitonin (PCT) has been proposed as a clinical parameter for the early detection of systemic bacterial infections and for guiding therapeutic management. The US Food and Drug Administration approved the biomarker PCT for the purpose of guiding antibiotic therapy in the context of acute respiratory infections and sepsis in February 2017 [1].

PCT consists of 116 amino acids and represents the precursor of the peptide calcitonin, from which it structurally and functionally stands out. Calcitonin acts by reducing blood calcium, opposing the effects of parathormones [6,7].

The CALC-1 gene, located on the short arm of chromosome 11, codes for a precursor peptide consisting of 141 amino acids. Immediately after its production, this pre-pro-hormone is degraded in its signal sequence 1–25 by a specific endopeptidase, generating PCT. This newly formed pro-hormone is then separated from specific proteolytic enzymes in N-PCT, calcitonin, and catacalcin, consisting of 57, 32, and 21 amino acids, respectively.

PCT is only present in small traces in circulation [2]. Its basal concentration in healthy patients is on the order of picograms at the limit of detectability, and most of the PCT is immediately converted into calcitonin [3,4].

However, in the case of conditions determining an over-expression of the CALC-1 gene, the plasma concentration of the PCT increases by thousands of times and remains unchanged, as it is unable to find blood enzymes capable of degrading it [5].

In physiological conditions, the expression of the CALC-1 gene is restricted to parafollicular cells of the thyroid gland. As for the “inflammatory” PCT, its production site is able to be found in all cells of neuroendocrine origin, encoded by the same CALC-1 gene [8].

Only a few authors have published results regarding the role of procalcitonin as a positive predictive factor in the early identification of complications of laryngeal surgery, such as the appearance of pharyngocutaneous fistulas. However, PCT values are known in relation to the early onset of fistulas in colorectal surgery or duodenal/pancreatic surgery [9]. Starting from the idea of the importance of PCT as a predictive index of infections [10], the aim of this study was to investigate the role of PCT as a predictive marker of early detection of pharyngocutaneous fistula (PCF) in a cohort of patients treated with total laryngectomy for laryngeal cancer (LC).

Pharyngocutaneous fistula is a serious complication after total laryngectomy, and it is associated with an increase in postoperative morbidity, a longer length of hospital stay, higher costs related to caregiving, and delaying oral feeding and adjuvant radio-chemotherapy. A pharyngocutaneous fistula occurs when there is a failure in neo-pharyngeal repair that results in salivary leaking. Its incidence after head and neck surgery varies, on average, from around 20% to 65% [11]. The large variety in the incidence of salivary fistulas can be explained by the tumor site, the previous treatment, and the technique of surgery or reconstruction [6].

There are multiple risk factors associated with PCF, yet it is difficult to predict the incidence of this complication in individual patients. Despite the increased understanding of risk factors for pharyngocutaneous fistulas and the improvements in surgical techniques, fistulas remain an important complication and occur without obvious cause in some patients with no known risk factors. A diagnosis of a fistula should be made as early as possible in order to reduce its associated morbidity [12].

## 2. Materials and Methods

A prospective, observational study was conducted on 43 consecutive patients who were enrolled at the U.O.C. of Otolaryngology of the A.O.U. Federico II of Naples from January 2019 to March 2022. All patients were informed regarding the methods, aims, and scope of the study.

The inclusion criteria were as follows: (1) histologically confirmed squamous cell carcinoma of the larynx; (2) patients who had never been treated before for laryngeal cancer (LC); (3) indication from our Multidisciplinary Team for upfront open surgical treatment according to the Head and Neck Cancers NCCN clinical practice guidelines [13]; and (4) informed consent.

The exclusion criteria were as follows: (1) patients who had undergone previous surgical or radio-chemotherapy treatment on the head and neck region; (2) patients with synchronous or metastatic tumors; (3) incomplete data records; (4) existence of preoperative infections and/or preoperative levels of PCT above 0.5 ng/mL; (5) less than one year of follow-up without evidence of disease; and (6) patients undergoing reconstruction of the pharyngo-esophageal tract using a flap.

The enrolled patients were subsequently divided as follows: group 1: laryngectomized patients with post-operative courses free from phlogistic and/or infectious complications; and group 2: laryngectomized patients with post-operative complications.

The surgical procedures were always performed by the same operating team, and the reconstructions of the pharyngo-esophageal tract were made using the same closure technique [14]. All patients were R0, with tumor-free margins of excision. Mono or bilateral neck dissections, according to site, size, and cN of the neoplasm, were performed. In all patients, the nasogastric feeding tube was attached during the surgery for nutrition. All of the patients were treated with antibiotic prophylaxis in a postoperative setting. Specifically, piperacillin/tazobactam or other penicillins, like ampicillin/sulbactam, were administered through intravenous therapy for 7 postoperative days [3].

All patients were subjected to a protocol of blood chemistry investigations, which was scheduled as follows: complete blood count with leukocyte formula, ESR dosage (mm/h), CRP (mg/L), and PCT (ng/mL). PCT was also dosed by salivary sampling and a pharyngo-cutaneous swab in patients who presented with PCF.

The dosage scheme was systematically repeated at time 0: day before intervention; time 1: 5th postoperative day; time 2: 20th postoperative day; and time X: appearance of the PCF (pertaining to the second group) between 4th and 11th post-operative days (median 7.3).

Procalcitonin was evaluated using a Vidas Brahms PCT (bioMerieux)^®^ semi-quantitative immunoassay. The biological samples (saliva and pharyngo-cutaneous swab), stored at −20 °C, were thawed at room temperature, and 200 microliters of each sample (an aliquot of saliva and liquid storage medium of the throat swab) were analyzed using the sandwich immunoenzymatic method, which is associated with a final fluorescence detection (ELFA—Vidas Brahms PCT^®^). The results were interpreted according to the following reference values: <0.09: negative, <0.09<: non-specific reactivity; =0.5: low positivity; >0.5–<2.00: medium positivity; >2.00–10.00: high positivity; >10.00: very high positivity with septic shock risk.

In terms of the cut-offs for positivity, other parameters were considered, such as ESR dosage with a value above 20 mm/h; CPR dosage with a value above 10 mg/L; and leukocyte values > 10.8 × 10^3^/mL at body temperature.

All patients were treated conservatively with compression dressings and, in the case of PCF, a switch to antibiotic therapy.

This study was conducted in accordance with relevant guidelines and regulations. It was approved by the institutional review board committee of the Federico II University of Naples, Naples, Italy (2022/207472).

## 3. Results

### Data Analysis

The 36 patients, with an average age of 68.41 years, were classified, according to the 8th edition of the TNM AJCC 2018 [15], as follows: 3 in stage II (pT2 N0), 11 in stage III (7 pT3 N0, 4 pT3 N1), and 22 in stage IVa (1 pT3 N2c, 12 pT4a N0, 6 pT4a N1, and 3 pT4a N2b).

All patients had positive histological examinations for laryngeal squamous cell carcinoma (SCC) with tumor-free margins of excision (R0).

A total of 27 patients, of which 21 were males and 6 were females, with an average age of 68.92 years, presented a course free from phlogistic and/or infectious complications; 9 patients, of which 7 were males and 2 were females, with an average age of 66.88 suffered PCF.

Table 1 and Table 2 summarize the clinical and oncological data. Protocol values for each group are reported in Table 3 and Table 4.

It should be noted that the seven patients with positive PCT values at time X had positive bacteriological swab for S. Aureus (5/7), Peptostreptococcus sp. (1/7), and Pseudomonas aeruginosa (1/7). No significant values or differences were found in the saliva sample.

Using the Mann–Whitney test, the concentrations of ESR and CPR were examined at times 0, 1, X, and 2, then compared between the two groups. An absence of statistical significance (*p* > 0.05) was found.

As regarding PTC, we observed that, for group 1 (without complications), the PCT was always 0 for T0, T1, and T2. Otherwise, in group 2, 100% of the sample had a value of 0 for PCT T0, while for T1, 77.8% had positive values of PCT (more than 0.5 ng/mL); for T2, 33.3% had a positive value; and for TX, 88.9% had a value of PCT greater than 0.5 ng/mL. To simplify the interpretation of the results, we have indicated negative PCT values with 0 and positive PCT values with 1 or higher than the laboratory reference range of normality.

Hence, for group 1, we did not need a test to observe the lack of a difference between T0, T1, and T2 for PCT. Alternatively, we tested whether there was a significant difference in the proportion of 1 in PCT among T0, T1, TX, and T2 for group 2. We used Cochran’s Q test (Figure 1).

The test was statistically significant (*p*-value < 0.001); hence, the PCT did not have the same proportion of 1 at all four time points. (test statistic 17,571; degree of freedom 3; asymptotic Sig. two-sided test < 0.001).

From the post hoc test, we demonstrate that the significant differences were between T0 and T1, T0 and TX, and T2 and TX in the PCF-positive group.

From the pairwise comparison of group 2 patients, we can see that there is a statistical significance difference between PCT T0 and PCT T1 (Std. test statistic = −3.240, *p*-value = 0.001), PCT T0 and PCT TX (Std. test statistic = −3.703, *p*-value < 0.001), and PCT T2 and PCT TX (Std. test statistic = −2.315, *p*-value = 0.021). (Table 5).

This result confirms that there is a statistically significant relationship between the PCT values found in group 2 patients at the various times in which it was dosed.

Once we understand this, we test if, for each time slot, there is a difference in the proportion of PCT value 1 between group 1 and group 2 using the independent sample proportion Z test.

For T0, PCT has the same proportion in both groups (0%). For T1, the proportion of positive PCT values in group 2 (77.8%) is greater than the proportion from group 1 (0%), and this is statistically significant (Z = −5.106, *p*-value < 0.001). For T2, in proportion of PCT value in group 2 (33.3%) is greater than the proportion in group 1 (0%), and this is statistically significant (Z = −3.133, *p*-value < 0.002). (Table 6)

## 4. Discussion

The impact of PCF fistulas on patient recovery and healthcare costs is notable [11].

Therefore, finding an early marker as an indirect index of this complication would be of fundamental help in establishing an adequate therapy. The most recent revalidations in the literature do not provide any indication regarding the use of PCT in this setting [2,9].

This study aimed to investigate the usefulness of procalcitonin as a marker for pharyngocutaneous fistulas after total laryngectomy, as demonstrated in different types of surgery, e.g., colorectal [16,17] or neurosurgery [18].

PCT is present in circulation only in small traces; its basal concentration in healthy patients is in the order of picograms, at the limit of detectability; most of this is immediately converted into calcitonin [3,4]. However, in the case of particular pathological conditions determining an over-expression of the CALC-1 gene, the plasma concentration of the PCT increases by thousands of times [2,8].

In particular, PCT production can be directly induced by bacterial toxins or indirectly by the cellular response to the host; it is precisely the combination of inflammatory cytokines and bacterial toxins that causes an increase in the transcription of the CALC-1 gene and, therefore, of circulating pro-hormone levels [11,19,20]. PCT release is mediated by cytokines that increase in response to bacterial infections, such as tumor necrosis factor-α, interleukin 1β, and interleukin 6, and is suppressed by interferons released in response to viral infections [7].

These results could be associated with disease severity and the clinical outcomes of patients with bacterial infections [21,22].

Currently, the emerging role of this marker is increasing: peak levels of PCT after a bacterial insult are usually achieved very rapidly, with values that are correlated with the intensity of stimulation [23]. PCT has a short half-life, and its levels usually drop rapidly after the end of the insult. For example, it is detectable within 2–4 h and peaks within 6–24 h, in contrast to CPR, which begins to rise after 12–24 h and peaks at 48 h. [20] Given this laboratory objectivity, the daily detection of the peptide could be adopted for post-operative patient monitoring [24].

Furthermore, unlike the reactive C protein, which is probably the most well-known and sensitive inflammatory marker, PCT shows a wide dynamic range in terms of states of infection. In fact, with the elimination of the infectious focus, the blood values of the peptide quickly fall within the physiological limits, unlike the CPR blood value, which can remain high days after the inflammatory focus has been eliminated [25].

In a recent meta-analysis, PCT was superior to C-reactive protein (CPR) in terms of differentiating bacterial from non-infectious causes of inflammation [26]. The other commonly used laboratory test used during the postoperative course for leucocyte count, VES, is neither very sensitive nor specific [27].

Regarding our data, we analyzed the PCT, ESR, and CPR values in patients undergoing total laryngectomy with and without infectious complications, represented by PCF. According to our data, there are no statistically significant differences between the ESR and CRP values in the two groups at the pre-established times.

The results obtained in the group of patients undergoing laryngeal surgery showed the significant relevance of PCT as a prognostic index of infection.

In particular, the peptide dosage was positive during the entire observation period in eight of nine patients who had postoperative pharyngeal–cutaneous fistulas.

Therefore, the finding of PCT positivization before clinical evidence of a fistula appears could make the peptide an important early marker of these complications.

The reduction in PCT levels at time 2 (20 days postoperative) could also be considered an indicator of the resolution of the illness.

In fact, it seems that the peptide under examination follows a trend directly mirrored by the evolution of the fistula, rising after surgery, maintaining positive values during the overt phase of infection, and finally decreasing around the twentieth day, which is the average duration of healing for a fistula. The reason that this occurs could be due to peri-operative contamination or bacterial translocation through saliva.

Similarly, authors have shown that PCT levels significantly increase in practically all patients after gastrointestinal surgery. They have also observed that PCT normalizes 5–7 days after surgery, but these levels remain high in cases of postoperative intra-abdominal infection [28].

Consequently, the dosage indications of the PCT in the perioperative setting of major head and neck surgery can be categorized into three main groups:Diagnosis of infections.

In healthy patients, the blood concentration of the PCT is kept below the dosage limits; an increase in blood concentration above 0.09 ng/mL could indicate an acute infection.

2.Control of therapy and the resolution of bacterial infections.

The hematic concentration of PCT can be a useful parameter for controlling the course of systemic bacterial infections and for monitoring the efficacy of therapy.

3.Monitoring and control of patients at risk.

The PCT assay can be used routinely in patients undergoing surgery who are at risk of a PCF.

Early detection allows for appropriate treatment (frequent medication, change or prolonged antibiotic therapy, and second surgery) and lower morbidity, in addition to a decreased need for re-operation and less deterioration in the patient’s quality of life [7,21,28].

## 5. Conclusions

Pharyngocutaneous fistulas are a serious complication after open laryngeal surgery, and they are determined by multiple risk factors. Despite our greater knowledge of the risk factors and improvements in surgical techniques, this is a complication that is difficult to predict.

PCT could be considered a specific early marker of complications. Based on the preliminary data obtained herein, where PCT [2,6,11,12,29,30] presented a close correlation with extra laryngeal infections and complications, it could be very useful in the early detection of PCF [9].

Further studies and multicentric data are encouraged in order to evaluate this biomarker, with the aim of decreasing antibiotic resistance and reducing healthcare costs.

## Figures and Tables

**Figure 1 cancers-16-00768-f001:**
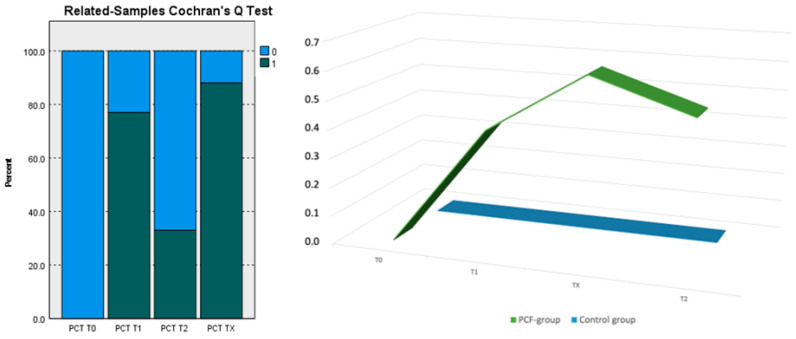
Distribution of PCT median values at T0; T1; Tx; T2 in the PCF-group (0; 0.42; 0.63; 0.53) and control group (0) *X* = time; *Y* = values.

**Table 1 cancers-16-00768-t001:** List of the 27 laryngectomized patients with post-operative courses free from phlogistic and/or infectious complications.

Patients	Patho Logies	pTNM	Subsite	Sex	Age
1	SCC	pT3N0	Glottis	F	69
2	SCC	pT4aN2b	Supraglottis	F	74
3	SCC	pT4aN2b	Glottis	F	55
4	SCC	pT4aN0	Glottis	F	63
5	SCC	pT3N0	Glottis	M	71
6	SCC	pT3N1	Glottis	M	77
7	SCC	pT2N0	Glottis	M	80
8	SCC	pT3N1	Glottis	M	64
9	SCC	pT4aN0	Glottis	M	68
10	SCC	pT4aN1	Glottis	M	67
11	SCC	pT4aN0	Glottis	M	59
12	SCC	pT3N1	Supraglottis	M	76
13	SCC	pT4aN0	Glottis	M	63
14	SCC	pT4aN1	Glottis	M	69
15	SCC	pT2N0	Glottis	M	71
16	SCC	pT4aN1	Glottis	M	77
17	SCC	pT4aN0	Glottis	M	72
18	SCC	pT3N0	Glottis	M	67
19	SCC	pT4aN0	Glottis	M	65
20	SCC	pT4aN0	Glottis	M	70
21	SCC	pT3N2c	Supraglottis	M	69
22	SCC	pT3N0	Glottis	M	72
23	SCC	pT4aN1	Glottis	F	69
24	SCC	pT4aN0	Glottis	M	70
25	SCC	pT3N0	Glottis	M	72
26	SCC	pT2N0	Glottis	M	65
27	SCC	pT3N1	Supraglottis	F	67

**Table 2 cancers-16-00768-t002:** List of the nine laryngectomized patients with post-operative complications.

Patients	Patho Logies	pTNM	Subsite	Sex	Age
1	SCC	pT4aN0	Glottis	F	69
2	SCC	pT4aN0	Subglottis	M	66
3	SCC	pT4aN1	Glottis	M	67
4	SCC	pT4aN0	Supraglottis	M	65
5	SCC	pT4aN2b	Glottis	M	68
6	SCC	pT3N0	Glottis	M	67
7	SCC	pT4aN0	Glottis	M	67
8	SCC	pT3N0	Glottis	F	66
9	SCC	pT4aN1	Glottis	M	67

**Table 3 cancers-16-00768-t003:** Parameters in patients without complications.

Group 1: 27 Patients			
	mT0	mT1	mT2
Body temperature	37.3 °C	37.3 °C	37.4° C
Leukocytosis	12.14 × 10^3^/mL	13.12 × 10^3^/mL	13.43 × 10^3^/mL
Positive ESR	18/27 patients: m = 34 mm/h (66.67%)	17/27 patients: m = 38 mm/h (74.07%)	13/27 patients: m = 28 mm/h (48.55%)
Positive CPR	15/27 patients: m = 30 mg/L (55.56%)	17/27 patients: m = 32 mg/L (62.96%)	15/27 patients: m = 22 mg/L (55.56%)
PCT in blood	negative	negative	Negative
PCT in saliva	negative	negative	negative

m: average value; T0: day before surgery; T1: five days after surgery; T2: twenty days after surgery.

**Table 4 cancers-16-00768-t004:** Infection parameters in patients with post-operative PCF.

Group 2: 9 Patients				
	mT0	mT1	mTX	mT2
Body temperature	37.2 °C	37.3 °C	37.7° C	37.3° C
Leukocytosis	12.35 × 10^3^/mL	12.62 × 10^3^/mL	12.78 × 10^3^/mL	12.64 × 10^3^/mL
Positive ESR	6/9 patients: m = 33 mm/h (66.67%)	7/9 patients: m = 34 mm/h (77.78%)	8/9 patients: m = 34.80 mm/h (88.89%)	6/9 patients: m = 29.50 mm/h (66.70%)
Positive CPR	4/9 patients: m = 21.30 mg/L (44.44%)	6/9 patients: m = 22.7 mg/L (66.67%)	7/9 patients: m = 26.80 mg/L (77.78%)	4/9 patients: m = 20.00 mg/L (44.44%)
PCT in blood and saliva	negative	6/9 patients: m = 0.42 ng/mL (77.78%)	8/9 patients: m = 0.63 ng/mL (66.67%)	4/9 patients: m = 0.52 ng/mL (44.44%)

m: average value; T0: day before surgery; T1: five days after surgery; T2: twenty days after surgery; TX: potential appearance of the PCF.

**Table 5 cancers-16-00768-t005:** Pairwise comparison: Each row tests the null hypothesis that the Sample 1 and Sample 2 distributions are the same. Asymptotic significances (two-sided tests) are displayed. The significance level is 0.50.

Sample 1–Sample 2	Test Statistic	Std. Error	Std. Test Statistic	Sig.
PCT T0-PCT T2	−0.333	0.240	−1.389	0.165
PCT T0-PCT T1	−0.778	0.240	−3.240	0.001
PCT T0-PCT TX	−0.889	0.240	−3.703	<0.001
PCT T2-PCT T1	0.444	0.240	1.852	0.064
PCT T2-PCT TX	−0.556	0.240	−2.315	0.021
PCT T1-PCT TX	−0.111	0.240	−0.463	0.643

**Table 6 cancers-16-00768-t006:** Independent sample proportion group statistics.

	Group	Successes	Trials	Proportion	Asymptotic Standard Error	Z	*p*-Value
PCT T0 = 1	=1	0	27	0.000	0.000	-	-
	=2	0	9	0.000	0.000	-	-
PCT T1 = 1	=1	0	27	0.000	0.000	−5.106	<0.001
	=2	7	9	0.778	0.139		
PCT T2 = 1	=1	0	27	0.000	0.000	−3.133	<0.002
	=2	3	9	0.333	0.157		

## Data Availability

Data are contained within the article.

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
