# Peer review of "Procalcitonin for Early Detection of Pharyngocutaneous Fistula after Total Laryngectomy: A Pilot Study"

_cancers, 2024, doi:10.3390/cancers16040768_

Round 1

Reviewer 1 Report (Previous Reviewer 1)

Comments and Suggestions for Authors

The manuscript is still of interest for each head and neck oncologic surgeon: this article explores a new technique which could detect the development of pharyngocutaneous fistula much earlier than standard approach.

This manuscript has been reviewed earlier and was rejected because the writing and presentation of the study was not according to standard international rules for scientific writing.

Compared to the rejected version, it has improved , but is still not acceptable for publication in its present form.

I again advise to take a course of science writing first, before writing an article.

Some suggestions to improve (It is not possible to re-write this article as a reviewer):

Title: Measurement of procalcitonin… instead of just Procalcitonin

Abstract:

Objective: … biomarker  .. : a biomarker of what ?? metastasis? I think the authors mean predictor of PCF , not a biomarker.

Do not present type of statistic test in the results section, this is part of the methods section. In the methofd section abstract: do not present location: it is not interesting for the reader to know where a study was conducted (in the abstract; such information of course has to be  presented in methods section).

Line 27-31 belongs in methods section, while 25-27 belongs in results section.

Conclusion: … predictor of complications in laryngeal surgery… : I guess not all complications (bleeding??) and not in all laryngeal surgery (removing papilloma of the larynx is also laryngeal surgery).

Introduction

In general: should be concise and should start form a general perspective and should end with the reasons and aim of the study formulating an hypothesis (which is tested in the study).

This is not the case in this article: so please re-write as indicated according to general scientific rules. For instance: line 55-69 could be transferred to the discussion section. Line 86-92 is not the hypothesis or the aim of the study, so this topic should be addressed earlier in the Introduction. Etcetera …

Materials and Methods:

Line 94 performed in  ..line 96: do you mean: informed consent?

Line 109-113: please give results in results section not in this section.

Line 122 CPR stands for cardiopulmonary resuscitation , do you mean C-reactive protein (CRP)?

Line 140: the authors tell that all (i.e. group 1 ánd group 2) patients were treated with compression dressings and switch of antibiotics, this means that also patients without signs of infection (group 1) were treated this way? If not, please re-read the manuscript carefully before submission.

Results:

Has improved, but not according to general rules. For instance, the reader has to calculate from table 1 and 2  the number sites, the number of T3 and T4, the number of N+? Please provide this information in your results section. Did all patients had neck dissection on both sides?

Line 161, 170, 173 and 187: do not introduce type of statistical test in the results section: this belongs in methods. Just describe results without drawing conclusions (which is part of the discussion section).

Line 164-167: what do the authors mean by PCT value of 1? In the methods section a valaue of 1 has been defined as only medium positivity (i.e. > 0.5 – 2.00), were there no PCT values> 1.00?

Tables:

-      legend Tabels do not describe the content of the Tables: for instance Table 3 could be: Infection parameters in patients without complications (group 1)

-      Normally not only mean, but also range, SD and median is presented in such Tables

-      Table 5 and 6: do not present raw statistical data or output from statistical software this way: describe the results in the text and tell the reader if a finding is statistically significant or not.

Figure 1: I see two figures: the legend does not describe the difference between them. There is no description of X-axis and Y-axis: how should the reader understand what he/she is looking at and could interpret these graphs?

Line 187: there is a difference: could the authors please describe what kind of difference?

Line 193: again, what is the definition of a PCT value of 1? Is that a concentration? Did I mis the definition?

Discussion:

In general, discussions are started with the most important finding of a study. Here it starts that the impact of PCF on health care costs is notable. But there is nothing in the results section indicating that health care costs have been analyzed in this study.  Use line 271-274 as your first Alinea, not the final topic of your discussion. Start with the most important discovery!

I completely agree with line 217, therefor it is of utmost importance to rewrite your article and make the head and surgeons in this world aware of this possible predictor of PCF. Please, please re-write your article in a much more scientific version.

Unfortunately, it is not doable to highlight every mistake in presentation and give suggestion for improvement for all sections. So please read your article carefully before submission.

Comments on the Quality of English Language

could be improved 

Author Response

the manuscript is still of interest for each head and neck oncologic surgeon: this article explores a new technique which could detect the development of pharyngocutaneous fistula much earlier than standard approach.

This manuscript has been reviewed earlier and was rejected because the writing and presentation of the study was not according to standard international rules for scientific writing.

Compared to the rejected version, it has improved , but is still not acceptable for publication in its present form.

I again advise to take a course of science writing first, before writing an article.

Some suggestions to improve (It is not possible to re-write this article as a reviewer):

Title: Measurement of procalcitonin… instead of just Procalcitonin

Abstract:

Objective: … biomarker  .. : a biomarker of what ?? metastasis? I think the authors mean predictor of PCF , not a biomarker.

Do not present type of statistic test in the results section, this is part of the methods section. In the methofd section abstract: do not present location: it is not interesting for the reader to know where a study was conducted (in the abstract; such information of course has to be  presented in methods section).

Line 27-31 belongs in methods section, while 25-27 belongs in results section.

  • Ok; I had tried to improve this section.

Conclusion: … predictor of complications in laryngeal surgery… : I guess not all complications (bleeding??) and not in all laryngeal surgery (removing papilloma of the larynx is also laryngeal surgery).

  • I had added in open laryngeal surgery.

Introduction

In general: should be concise and should start form a general perspective and should end with the reasons and aim of the study formulating an hypothesis (which is tested in the study).

This is not the case in this article: so please re-write as indicated according to general scientific rules. For instance: line 55-69 could be transferred to the discussion section. Line 86-92 is not the hypothesis or the aim of the study, so this topic should be addressed earlier in the Introduction. Etcetera …

  • the cited section has been reformulated according to the suggested indications.

Materials and Methods:

Line 94 performed in  ..line 96: do you mean: informed consent?

  • yes

Line 109-113: please give results in results section not in this section.

  • Ok

Line 122 CPR stands for cardiopulmonary resuscitation , do you mean C-reactive protein (CRP)?

  • correct

Line 140: the authors tell that all (i.e. group 1 ánd group 2) patients were treated with compression dressings and switch of antibiotics, this means that also patients without signs of infection (group 1) were treated this way? If not, please re-read the manuscript carefully before submission.

  • Yes the patients were treated the same way.

Results:

Has improved, but not according to general rules. For instance, the reader has to calculate from table 1 and 2  the number sites, the number of T3 and T4, the number of N+? Please provide this information in your results section. Did all patients had neck dissection on both sides?

  • the patients were all treated with bilateral neck dissection (II-IV). The number of patients per stage is reported (line 145-147).

Line 161, 170, 173 and 187: do not introduce type of statistical test in the results section: this belongs in methods. Just describe results without drawing conclusions (which is part of the discussion section).

  • ok

Line 164-167: what do the authors mean by PCT value of 1? In the methods section a valaue of 1 has been defined as only medium positivity (i.e. > 0.5 – 2.00), were there no PCT values> 1.00?

  • In the methods section, the cut-off value to define the positivity of PCT values was defined as 0.5 - 2.0.
  • In order to avoid reporting the decimals for each individual patient in whom the PCT dosage was carried out, we simplified it by saying that in the group there were taken into consideration values > 1 (therefore positivity) for patients at the TX time; T1; T2; reporting the percentages for each value of T.

Tables:

-      legend Tabels do not describe the content of the Tables: for instance Table 3 could be: Infection parameters in patients without complications (group 1)

- ok

-      Normally not only mean, but also range, SD and median is presented in such Tables

- ok

-      Table 5 and 6: do not present raw statistical data or output from statistical software this way: describe the results in the text and tell the reader if a finding is statistically significant or not.

-     correct

Figure 1: I see two figures: the legend does not describe the difference between them. There is no description of X-axis and Y-axis: how should the reader understand what he/she is looking at and could interpret these graphs?

  • Ok, I had correct

Line 187: there is a difference: could the authors please describe what kind of difference?

  • The note in figure 1 has been corrected: the difference lies in the distribution of PCT values. in group 1 they are always equal to zero; for group 2 the curve shown in the figure is described instead.

Line 193: again, what is the definition of a PCT value of 1? Is that a concentration? Did I mis the definition?

  • It is a value to define positivity.

Discussion:

In general, discussions are started with the most important finding of a study. Here it starts that the impact of PCF on health care costs is notable. But there is nothing in the results section indicating that health care costs have been analyzed in this study.  Use line 271-274 as your first Alinea, not the final topic of your discussion. Start with the most important discovery!

I completely agree with line 217, therefor it is of utmost importance to rewrite your article and make the head and surgeons in this world aware of this possible predictor of PCF. Please, please re-write your article in a much more scientific version.

Unfortunately, it is not doable to highlight every mistake in presentation and give suggestion for improvement for all sections. So please read your article carefully before submission.

  • I tried to improve the section according to the indications received.

Reviewer 2 Report (Previous Reviewer 2)

Comments and Suggestions for Authors

This manuscript is improved from the initial draft. The authors have addressed many of the concerns raised. Please avoid using abbreviations without first spelling out the abbreviated phrase (eg ESR, CPR). The figures and data analysis is much more clear.

Comments on the Quality of English Language

The paper still needs a great deal of revision from an English language perspective. 

Author Response

Dear Reviewer,
I thank you for the additional time you have dedicated to reading this work, which has improved in many parts.
The English had been corrected by the magazine itself complete with a translation certificate. If you think there is still work to do, please let the publisher know as we have used the official English editing service.

This manuscript is a resubmission of an earlier submission. The following is a list of the peer review reports and author responses from that submission.

Round 1

Reviewer 1 Report

Comments and Suggestions for Authors

This study is an interesting attempt to validate procalcitonin as a predictive biomarker for pharyngocutaneous fistula after total laryngectomy. This approach is new and should be published in my opinion.

However the present form is not acceptable and should be changed. I have major and serious concerns.

1. there are far too many mistakes in English language (spelling) and no consistent use of abbreviations .

2. the manuscript does not fulfill the rules of scientific writing:

a. introduction is not concise

b. result and discussion section: lot of repetition of sentences from introduction / methods part

c.  results should not be presented as "raw" data in tables. Please read other reports as an example how to avoid this. It looks like a complete spread sheet of all the available data is copied into a table .... unbelievable... it almost stopped me reviewing.

d. results which can be presented in tables should not be written down (line 147-184 can be presented as a table and therefore easier to read and to interpret).

e. it is not done, not necessary and ethically completely wrong to present initials of patients in a scientific paper !! 

f. Discussion: again not according to rules of scientific publishing: results of this study should be discussed in relation to existing literature. Some sentences are almost copied from the introducton (line 105-11 could be removed for example).

In detail (numbers refer to line):

16 procalcytoin ?

17 all post surgical complications or wound complications?

22 no results are shown regarding neck abscess; did they dissapear (also line 75)?

27 why protocol between quotation marks?

28 what does "with formula" mean? ESR not explained, a dosage is administration of a medicine (also line 118), do the authors mean assessed/measured? abbreviation PCT not explained, dosage scheme (measurements?)

33 in /at the time?

34 p value alone is not interesting: please provide (mean/median) PCT concentrations of time points.

36 conclusion regarding sensitivity and specificity without providing any value of these parameters in the results. what was positive/negative predictive value? Accuracy? 

78 increate?

79 adjuvant radiotherapy?

80 whwn?? repair? leak(age)?

81 haed?

84 poor prognosis or predictor? there is a fundamental difference between predictive and prognostic value.

88 why do the authors introduce PCF and do not use this abbreviation (the same for other abbreviations used)

96 ".." after study

98 LC?

99 what is a disease management team: do the authors mean multidisciplinary team?

100 just: give informed consent

101 1 year follow-up can never be an inclusion criteria: you do not know before inclusion if the patient will have 1 year FU. It can be an exclusion criterion

104 what is the definition of incomplete data?

111 how was the pharyngo-esophageal tract reconstructed: in case of flaps please remove these patients form analysis (unless alle patients had a flap)

112 nasogastric tube

114 in the first sentence of the introduction the authors mention a reason for this study was to avoid antibiotic prescriptions, but this study is performed in a patient group using standard antibiotics. The authors should explain in the discussion section why.

137 why again informed consent?

142 repetition: groups are already defined in methods section

147-184 please summarize in table and remove table with raw data per patient.

188 this section is actually results. in the results section you present the main results after statisctal analysis. You do not describe the statistics itself 9so; remove title sub-section 3.2)

line numbers restart after table?

discussion line 

2-4 repetition: please remove

19 the same

a lot of information in the Discussion section without reflecting your own study results.

43 PCR??

60 finally: regarding our data. this should be done eralier. start you discussion section with the most important results and reflect this to the existing literature . and so on ..

90- 100: quite jumpy: did you study these head and neck groups and can you draw these conclusion or is it just wishful thinking? 

105-112: repetition of introduction. please keep conclusions concise.

189 repetition: mann-Whithey

Comments on the Quality of English Language

already described above

Author Response

REVIEWER 1

This study is an interesting attempt to validate procalcitonin as a predictive biomarker for pharyngocutaneous fistula after total laryngectomy. This approach is new and should be published in my opinion.

However the present form is not acceptable and should be changed. I have major and serious concerns.

  1. there are far too many mistakes in English language (spelling) and no consistent use of abbreviations .
    1. We will edit in English.
    2. the manuscript does not fulfill the rules of scientific writing:
    3. introduction is not concise
    4. result and discussion section: lot of repetition of sentences from introduction / methods part
    5. results should not be presented as "raw" data in tables. Please read other reports as an example how to avoid this. It looks like a complete spread sheet of all the available data is copied into a table .... unbelievable... it almost stopped me reviewing.
    6. results which can be presented in tables should not be written down (line 147-184 can be presented as a table and therefore easier to read and to interpret).
    7. it is not done, not necessary and ethically completely wrong to present initials of patients in a scientific paper !! 
    8. Discussion: again not according to rules of scientific publishing: results of this study should be discussed in relation to existing literature. Some sentences are almost copied from the introducton (line 105-11 could be removed for example).

In detail (numbers refer to line):

16 procalcytoin ?

  • Correct in procalcitonin

17 all post surgical complications or wound complications?

  • Correct: post surgical wound complication

22 no results are shown regarding neck abscess; did they dissapear (also line 75)?

  • Our intention is to evaluate the appearance of pharynx-cutaneous fistula and the relationship with procalcitonin levels. We did not analyze conditions such as neck abscess also because they did not occur.

27 why protocol between quotation marks?

  • correct

28 what does "with formula" mean? ESR not explained, a dosage is administration of a medicine (also line 118), do the authors mean assessed/measured? abbreviation PCT not explained, dosage scheme (measurements?)

  • leukocyte formula; PCT is short for procalcitonin as explained on line 50. I entered the units of measurement for ESR; PCR, PCT.

33 in /at the time?

  • In time

34 p value alone is not interesting: please provide (mean/median) PCT concentrations of time points.

-I have correct completely statistical analysys.

36 conclusion regarding sensitivity and specificity without providing any value of these parameters in the results. what was positive/negative predictive value? Accuracy? 

  • correct

78 increate?

  • correct

79 adjuvant radiotherapy?

  • correct

80 whwn?? repair? leak(age)?

  • correct

81 haed?

  • correct

84 poor prognosis or predictor? there is a fundamental difference between predictive and prognostic value.

  • Pharyngocutaneous fistula is an unfavorable prognostic factor for the patient as it can lead to more serious complications such as rupture of the carotid artery.

88 why do the authors introduce PCF and do not use this abbreviation (the same for other abbreviations used)

  • correct

96 ".." after study

  • correct

98 LC?

  • Laryngeal cancer

99 what is a disease management team: do the authors mean multidisciplinary team?

  • Multidisciplinary team

100 just: give informed consent

  • correct

101 1 year follow-up can never be an inclusion criteria: you do not know before inclusion if the patient will have 1 year FU. It can be an exclusion criterion

  • correct

104 what is the definition of incomplete data?

  • Failure to report the onset of the complication (e.g. day of appearance of the fistula), failure to carry out pre-established tests within the pre-established times, alteration in data collection (e.g. blood sample damaged during transport to the laboratory)

111 how was the pharyngo-esophageal tract reconstructed: in case of flaps please remove these patients form analysis (unless alle patients had a flap)

  • I included patients who underwent flap reconstruction in the exclusion criteria.

112 nasogastric tube

  • correct

114 in the first sentence of the introduction the authors mention a reason for this study was to avoid antibiotic prescriptions, but this study is performed in a patient group using standard antibiotics. The authors should explain in the discussion section why.

  • The aim of our study is not to evaluate the rapid resolution of complications after improved antibiotic therapy but to follow the trend of procalcitonin in the immediate period preceding the onset of the complication, during the complication and the release of procalcitonin levels in the phase of resolution. In our study, the pathogens found as infectious agents proved to be sensitive to standard antibiotic therapy. A possible objective of a future study will be to evaluate the rapid resolution of procalcitonin levels or the rapid resolution of the complication after changing the antibiotic therapy used. Introducing this variable would have altered the results of the study.

137 why again informed consent?

  • correct

142 repetition: groups are already defined in methods section

  • I moved what was written to the line 107 methods section

147-184 please summarize in table and remove table with raw data per patient.

  • I corrected tables 1 and 2 and inserted the data from the materials and methods section into tables 3 and 4.

188 this section is actually results. in the results section you present the main results after statisctal analysis. You do not describe the statistics itself 9so; remove title sub-section 3.2)

  • ok

line numbers restart after table?

  • the numbering problem is one of layout. the editor inserted the tables and the altered numbering came out.

discussion line 

2-4 repetition: please remove

  • Ok

19 the same

  • OK

a lot of information in the Discussion section without reflecting your own study results.

43 PCR??

  • YES: CPR

60 finally: regarding our data. this should be done eralier. start you discussion section with the most important results and reflect this to the existing literature . and so on ..

90- 100: quite jumpy: did you study these head and neck groups and can you draw these conclusion or is it just wishful thinking? 

  • Si tratta di prospettive future sulla base della nostra esperienza se pur piccola.

105-112: repetition of introduction. please keep conclusions concise.

  • ok

189 repetition: mann-Whithey

Reviewer 2 Report

Comments and Suggestions for Authors

This is an interesting case series evaluating the potential utility of procalcitonin levels for the detection of pharyngocutaneous after total laryngectomy for laryngeal cancer. 

Please clarify:

- was procalcitonin drawn from blood, patient saliva or neck wound. Methods and Results need to be made more clear

- what is PCR. This abbreviation is used throughout the manuscript with no description of what this is

- Where any flaps used in reconstruction? If no, please state in methods

- Did detection of elevated procalcitonin levels in this cohort modify treatment plans?

- It should be pointed out in the discussion that a 7 week course of iv antibiotics after TL is not currently supported by the literature

- Where drug resistant organisms isolated from the PCF cohort?

- how long were patients made npo post op

- was there any documented emesis in TL patients? this being a risk factor for PCF

- do the authors order a pharyngogram-esophogram post op? at what day post op? When and how is PO resumed?

- describe the treatment of PCFs in this cohort: conservative wound care/antibiotics? surgical exploration? flap?

A figure showing the differences in PCT levels at different time points between the 2 groups would be helpful.

Overall, while this manuscript requires significant edits for clarification and full elucidation of methodology, the findings are novel within our field of interest. 

Comments on the Quality of English Language

Please have a native English speaker edit the manuscript for grammar, vocabulary, fluidity and sentence structure. 

Author Response

Reviewer 2

This is an interesting case series evaluating the potential utility of procalcitonin levels for the detection of pharyngocutaneous after total laryngectomy for laryngeal cancer. 

Please clarify:

- was procalcitonin drawn from blood, patient saliva or neck wound. Methods and Results need to be made more clear

- I specified for each individual patient where the biological material was collected. I have also added summary tables that clarify things better.  Procalcitonin levels were studied on a blood sample; salivary collection; wound swab

- what is PCR. This abbreviation is used throughout the manuscript with no description of what this is

- CRP: C REACTIVE PROTEIN

- Where any flaps used in reconstruction? If no, please state in methods

- No flaps were used. The complication was treated with conservative therapy and antibiotic therapy.

- Did detection of elevated procalcitonin levels in this cohort modify treatment plans?

- The treatment was not changed as the aim of our study was only to evaluate the proalcitonin values in relation to the onset and resolution of the pharyngocutaneous fistula. The introduction of other types of medical therapy could have distorted the data obtained.

- It should be pointed out in the discussion that a 7 week course of iv antibiotics after TL is not currently supported by the literature

- It is 7 days not 7 week. The reference is numer 3

- Where drug resistant organisms isolated from the PCF cohort?

- no

- how long were patients made npo post op

-  Until the fistula resolves

- was there any documented emesis in TL patients? this being a risk factor for PCF

- No one

- do the authors order a pharyngogram-esophogram post op? at what day post op? When and how is PO resumed?

- It was not performed in any patient. this is certainly a limit but it is a mere organizational question.

- describe the treatment of PCFs in this cohort: conservative wound care/antibiotics? surgical exploration? flap?

- All patients were treated conservatively with compression dressings and antibiotic therapy (piperacillin/tazobactam or other penicillins like ampicillin/sulbactam) until the fistula resolves.

A figure showing the differences in PCT levels at different time points between the 2 groups would be helpful.

  • I have added table 3 and 4

Overall, while this manuscript requires significant edits for clarification and full elucidation of methodology, the findings are novel within our field of interest. 

  • I used mdpi editing

Round 2

Reviewer 1 Report

Comments and Suggestions for Authors

After reading carefully the changes made by the authors I have to conclude the my major concerns were not taken into account en the manuscript did not improve. The introduction in still not concise and far to detailled. Results section now contains 14 (!) Tables of which many with raw statistical results. Even italian language in a Table. Still the raw data of the patients instead of a normally presented Table 1 with patient characteristics. 

Although the concept of the study is original and of potentially clinical/ scientific interest, the scientific presentation is far below what has to be expected in a peer-reviewed journal: this is not the way a scientific paper has to be written. 

Comments on the Quality of English Language

there are still Italian words in the text and the English language is still not right

Author Response

After reading carefully the changes made by the authors I have to conclude the my major concerns were not taken into account en the manuscript did not improve. The introduction in still not concise and far to detailled. Results section now contains 14 (!) Tables of which many with raw statistical results. Even italian language in a Table. Still the raw data of the patients instead of a normally presented Table 1 with patient characteristics. 

Although the concept of the study is original and of potentially clinical/ scientific interest, the scientific presentation is far below what has to be expected in a peer-reviewed journal: this is not the way a scientific paper has to be written. 

I have made the previously requested changes. We believe we should leave the introduction intact to explain the role of the PCT, its composition and the aims of the study.

Tables 1 and 2 include the patients enrolled in the study. For each table the age, sex, staging (TNM) and primary site of the tumor were reported. I was subsequently asked to report the data presented in the results section in tables to make interpretation of the data easier. I have therefore reformulated new tables and inserted the tables of the statistical part which has been completely revised. I reduced the number of tables

I don't understand what English language errors are present given that the work, before being uploaded, was completely revised by the editing service of the MDPI magazine which also corrected the tables in their language.

If it helps, I can reload the work without highlighted corrections to make reading easier.

Reviewer 2 Report

Comments and Suggestions for Authors

Authors need to acknowledge the following paper: https://pubmed.ncbi.nlm.nih.gov/37060781/

The authors fail to identify what they are measuring by PCR. This is never described in the paper.

The figures/tables need to be simplified and made simpler. The legends need to be more descriptive and complete in order to better understand what the authors are presenting in the figures/tables. 

Comments on the Quality of English Language

need some improvement still. 

Author Response

Authors need to acknowledge the following paper: https://pubmed.ncbi.nlm.nih.gov/37060781/

  • I inserted the suggested study. At the time of writing the work had not yet been published. We can say that we reached similar but completely independent conclusions in a short space of time. LINE 73; REF 9

The authors fail to identify what they are measuring by PCR. This is never described in the paper.

  • I apologize for the inaccuracy. PCR was correctly translated into CPR (c-reactive protein). The relationship between CPR and infection and the comparison between CPR and PCT are explained in:Line 142; Table 3; Pag 12 line 37 - 51

The figures/tables need to be simplified and made simpler. The legends need to be more descriptive and complete in order to better understand what the authors are presenting in the figures/tables. 

  • I have corrected the table legends to make them easier to understand. I reduced the number of tables.

I'm sorry that once again you tell me to review my English. The text was corrected by the editing service of the MDPI magazine complete with a certificate of language correction. I deliberately left the corrections even if it is more difficult to read in order to highlight them. If you want I can upload the corrected file without deletions to make reading easier.

Round 3

Reviewer 1 Report

Comments and Suggestions for Authors

The manuscript is still of potential interest for each head and neck oncologic surgeon, however the scientific presentation is not acceptable as stated earlier. Introducing even more tables (as answer to my serious concerns) with raw statistics and raw data is not the way a scientific report has to be written. 

Just some examples:

- in abstract the suggestion has been made that this study also evaluates neck abscesses (line 22), but this was not the case (as mentioned earlier)

- introduction: still far too long and referring to literature without references ( line 91-96) : if you tell the reader that there multiple risk factors you have to provide the literature where you found this (It is not allowed to give you own opinion in an Introduction section).  

- the same holds for line 46 ( association with higher costs, etc: is that you r own conclusion or is there evidence and if so: which study?); line 73 and 86: only a few authors .. (and only 1 reference ???) ; line 86: ... the large variety can be explained .. is that your own opinion (presented as a fact which is not allowed) or is this a conclusion drawn form the literature (if so: please present reference).

- The intruduction still not straight forward : line 51-72 is not interesting for a clinician and is not elaborated in the results or methods, so you can remove this and and just refer to the literature describing it. 

In methods sectiion: a lot is missing for instance: which statisctical program has been used and which test have been used? This is sometimes provided in teh results section where it does not belong.

The results section is not presented properly: it is still very chaotic. You are NOT summarizing data (in Table 1 or 2) but just present raw data! The interested reader has to figure out him/herself what the mean age is, the number of T2 or T4 , stage etc etc ... It is really not the way results have to be presented in a peer reviewed journal . PLEASE read other scientific acticles to understand this.

In a results section (this is common knowledge for scientific writers) a summary of results has to be presented either in text or in tables. Not both in text as well as in tables. Do not present raw statistical information (copied from the output of statistical software) in Tables.... PLEASE read scientific articles. Reduce number of tables give summaries instead of raw data. What does Tab mean? Table 6 is also a figure and it does not provide any additional information: it is just raw statistical output confusing the interested reader.

The same holds for Table 7-10: please do NOT present results in this way! 

Results line 161- 191: does the reader compare each value by him/herself? why is it presented in this way? please read other scientific articles: results comparing groups do not have to be presented this way...

General advice: please follow a science writing course before submitting a manuscript. Do not get me wrong: the scientific idea and the analysis itself is good and interesting, but the writing and presentation is unfortunately not according to general rules of scientific writing.

Comments on the Quality of English Language

no additional comments

Author Response

The manuscript is still of potential interest for each head and neck oncologic surgeon, however the scientific presentation is not acceptable as stated earlier. Introducing even more tables (as answer to my serious concerns) with raw statistics and raw data is not the way a scientific report has to be written. 

Just some examples:

- in abstract the suggestion has been made that this study also evaluates neck abscesses (line 22), but this was not the case (as mentioned earlier)

-           I Have removed

- introduction: still far too long and referring to literature without references ( line 91-96) : if you tell the reader that there multiple risk factors you have to provide the literature where you found this (It is not allowed to give you own opinion in an Introduction section).  

- The reference to lines 91-96 is indicated and corresponds to reference 12

- the same holds for line 46 ( association with higher costs, etc: is that you r own conclusion or is there evidence and if so: which study?); line 73 and 86: only a few authors .. (and only 1 reference ???) ; line 86: ... the large variety can be explained .. is that your own opinion (presented as a fact which is not allowed) or is this a conclusion drawn form the literature (if so: please present reference).

- All references have been corrected with scientific evidence from the literature.

- The intruduction still not straight forward : line 51-72 is not interesting for a clinician and is not elaborated in the results or methods, so you can remove this and and just refer to the literature describing it. 

- I believe that the lines cited are necessary to explain the derivation of procalcitonin and its metabolism. Also in this case the bibliographical references are correct.

In methods sectiion: a lot is missing for instance: which statisctical program has been used and which test have been used? This is sometimes provided in teh results section where it does not belong.

  • Correct

The results section is not presented properly: it is still very chaotic. You are NOT summarizing data (in Table 1 or 2) but just present raw data! The interested reader has to figure out him/herself what the mean age is, the number of T2 or T4 , stage etc etc ... It is really not the way results have to be presented in a peer reviewed journal . PLEASE read other scientific acticles to understand this.

  •  

In a results section (this is common knowledge for scientific writers) a summary of results has to be presented either in text or in tables. Not both in text as well as in tables. Do not present raw statistical information (copied from the output of statistical software) in Tables.... PLEASE read scientific articles. Reduce number of tables give summaries instead of raw data. What does Tab mean? Table 6 is also a figure and it does not provide any additional information: it is just raw statistical output confusing the interested reader.

  • The results section was subjected to correction by a specific statistics company that collaborates with important scientific journals. for this reason we believe both the results and the tables are clear, as the other reviewers have also highlighted.

The same holds for Table 7-10: please do NOT present results in this way! 

  • The same response

Results line 161- 191: does the reader compare each value by him/herself? why is it presented in this way? please read other scientific articles: results comparing groups do not have to be presented this way...

  • This is a comparison between the values taken into analysis that correlate with an infection situation.

General advice: please follow a science writing course before submitting a manuscript. Do not get me wrong: the scientific idea and the analysis itself is good and interesting, but the writing and presentation is unfortunately not according to general rules of scientific writing.

Round 4

Reviewer 1 Report

Comments and Suggestions for Authors

I would like to thank the authors for their attempts to improve this article. Still the results section has not been improved. Although their is a start of scientifically presentation of results (line 249-259), the tables and figures do not give information to a head and neck clinician. It is still a copy of raw statistics. the fact that statistical analysis has been performed by an experienced, highly regarded individual or company does not relieve the authors of their responsibility to analyze these statistical results and present them in the results section in a manner that is customary and easy for clinicians to interpret. Although I understand the tables through my years of experience in medical scientific research, this is not the way to present it in a clinically oriented journal. In addition, conclusions are now even drawn in the results section, which, as is known, does not belong in a scientifically designed article. not to mention that the order and abbreviations of many of these tables and figures are no longer displayed correctly.

Comments on the Quality of English Language

has been improved